# Dynamics Modeling and Analysis of Rolling Bearings Variable Stiffness System with Local Faults

Baoliang Guo, Wenlong Wu, Jianxiao Zheng *, Yumin He and Jinhua Zhang

School of Mechanical and Electrical Engineering, Xi'an University of Architecture and Technology, Xi'an 710055, China; gbl93@xauat.edu.cn (B.G.); 995@xauat.edu.cn (W.W.); heyumin@xauat.edu.cn (Y.H.); zhangjh0808@xauat.edu.cn (J.Z.)
* Correspondence: zjx@xauat.edu.cn; Tel.: +86-130-729-12676

**Abstract:** By analyzing the support of load-carrying rolling elements when the rolling elements fall into the fault position, the dynamics model of a rolling bearing variable stiffness system with local faults is proposed, considering the retention factor of the contact deformation. Then, this paper researches the change of effective contact stiffness, contact deformation, contact force, and the total effective stiffness of the rolling elements. The results show that the contact stiffness of the rolling elements abruptly decreases when the rolling elements fall into the fault position. The contact deformation and contact force of the load-carrying rolling elements in the load zone increase, rebalancing the external radial load while causing a sudden reduction in the total effective stiffness, resulting in the vibration of the system. When different rolling elements fall into the outer ring fault position, the change in total effective stiffness and the system response are equal in magnitude. Additionally, there is a significant outer race fault characteristic frequency accompanied by frequency multiplication in the fault characteristic spectrums. When different rolling elements fall into the inner race fault position, the total effective stiffness is modulated by the inner race rotation and varies dramatically, resulting in the amplitude of the system time domain vibration response also being modulated by the inner race rotation and varying dramatically. Additionally, there is a significant inner race rotational frequency accompanied by frequency multiplication, an inner race fault characteristic frequency accompanied by frequency multiplication, and a side frequency in the fault characteristic spectrums. The research can provide some reference for the effective diagnosis of the rolling bearing fault.

**Keywords:** rolling bearings; local faults; dynamics; variable stiffness; fault diagnosis

## 1. Introduction

With the continuous advancement of industrial technology, rotating machinery is developing in the direction of automation, integration, intelligence, high speed, and precision, which has led to an increase in the fault rate. As a key component widely used in rotating machinery, rolling bearing is prone to failure under harsh working conditions [1]. The unexpected fault of bearings can lead to a sudden collapse of a machine or system, and it may result in significant economic losses or even casualties [2]. Bearing fault diagnosis is mainly performed in two aspects; on the one hand, the fault information is extracted from the vibration response signal, which is mainly extracted and diagnosed by various learning algorithms. For example, the deep learning approach [3,4], digital twin-driven approach [5], optimized adaptive deep belief network [6], etc. [7,8]. With the increase in the use of smart machines, the detection and diagnosis of mechanical faults by these methods are increasing day by day. On the other hand, bearing dynamics models are mainly studied. Rolling bearing fault dynamics is used as a comprehensive and accurate method to predict the vibration characteristics of rolling bearings with various faults and to provide in-depth guidance for detection and diagnosis applications [9]. Fault diagnosis mechanisms and

methods for ball bearings are an ongoing research focus [10]. The dynamic models of shaft bearing systems have been developed for theoretical studies [9,11].

Cui et al. [12–16] developed a rolling bearing dynamics model by introducing the circumferential and radial dimensional parameters of two-dimensional faults into the displacement excitation function, estimated the magnitude of the faults by analyzing the time interval characteristics of the impact response, predicted the effect of local defects on ball bearing vibration, and also studied the vibration of deep groove ball bearings with single and multiple defects on the inner and outer race surfaces. Wu et al. [17–20] described the geometric displacement of the rolling body through a two-dimensional fault with a segmentation function and used this displacement as displacement excitation to establish a rolling bearing dynamics model, observed the relationship between vibration response and fault size, considered the coupling of rolling elements and segmentation effects, brought the acceleration response of the model more in line with the actual situation, and also considered the influence of the dent shoulder on the vibration, and derived a more close to real impulse characteristics caused by a practical dent. To build the rolling bearing dynamics model, Qui et al. [21,22] use the segmented function to introduce the three-dimensional geometric parameters of the fault into the displacement excitation. They analyze the variation of contact force of rolling elements under different contact types and the relationships between the fault size and the system vibration response. Zhang et al. [23,24] considered the dynamic lubrication conditions in an elastic fluid, described the geometric displacement of a rolling body as it passes through a fault using a segmentation function, used this displacement as a displacement excitation, developed a rolling bearing dynamics model, introduced the transient collision force excited by the strike of the rolling element on the trailing edge of the spall area, and analyzed the double pulse time interval, and through additional deflection and multi-impact theories, it was found that the location and the number of impulses due to varying compliance strongly depend on multiple factors, and mainly on the values of applied load and shaft rotational speed. Gao et al. [25,26] coupled displacement-excited rolling bearings in rail vehicles and rotor systems, respectively. Petersen et al. [27–29] established the system dynamics equations based on the displacement excitation function and analyzed the characteristics of the system stiffness and contact force change under different fault dimensions, i.e., the stiffness decreases in the loading direction and increases in the unloading direction, and also proposed a method for accounting for the finite rolling element size, which means that the time-frequency characteristics of the low-frequency event that occurs when a rolling element enters the defect entry and the multiple high-frequency events that occur when it exits the defect can be predicted more accurately. Sarabjeet et al. [30] used the explicit dynamics finite element software package LS-DYNA to build a rolling bearing fault model, and after an in-depth analysis of the numerically estimated dynamic contact forces between the rolling elements and the raceways of a bearing, it was found that the re-stressing of the rolling elements that occurs near the end of a raceway defect generates a burst of multiple short-duration force impulses, and the contact forces and accelerations generated on the exit of the rolling elements out of the defect compared to when they strike the defective surface are much higher.

The above-mentioned rolling bearing faulty dynamic models can be used in the analysis and diagnosis of system faults. In these dynamic models, the maximum displacement of the impact excitation is calculated by the fault size and the bearing geometric parameters. Additionally, the contact deformation and force are calculated by the displacement when the rolling elements fall into the fault position. Because of the rigid nature of the rolling elements, the deformation is generally a low order of magnitude. The maximum displacement is far greater than the contact deformation, so there is a large deviation.

In general, the radial load of the bearing is balanced by the contact force of the multiple load-bearing rolling elements. When a rolling element falls into the fault position and loses all or part of its load-carrying capacity, the radial load is redistributed between the load-bearing rolling elements. Additionally, the contact deformation and contact force of

each load-bearing rolling element will be increased. When the rolling element loses its load capacity, that is, its contact stiffness, the total contact stiffness of the system will be changed suddenly. Then, the shock of the system is caused.

Based on Hertz's contact theory, the retention factor of contact deformation will be defined in this paper. During the rotation of different rolling elements with the cage, this paper analyzes the variation of the effective contact stiffness, contact force, and total effective stiffness of the system in the direction of radial load. A dynamic model of a single degree of freedom variable stiffness of a rolling bearing system with a single local damage fault is established, and simulation analysis and experimental verification are carried out.

## 2. Establish a Dynamic Model of the System with Fault

The deep groove ball bearing structure under radial load is shown in Figure 1a, the numbers in the figure are rolling element serial numbers, simplifying the elastic contact of a single rolling element with the inner and outer races into springs and dampers, as shown in Figure 1b. In Figure 1, $O_o$ and $O_i$ are the centers of the outer and inner races, $\beta_0$ is the initial position angle of the fault location, $\psi$ is the azimuth at which the rolling elements are located, $Q_{max}$ is the maximum contact force in the load zone, $f_m$ is the cage rotation frequency, $f_s$ is the inner race rotation frequency, $F_r$ is an external radial load, the amplitude and direction are fixed, and $\delta_r$ is the maximum relative displacement between the inner and outer races. Assume that the $x$-axis is in the radial load direction and the positive direction is the same as the radial load direction and the sensor is set on the outer race in the positive direction of the $x$-axis.

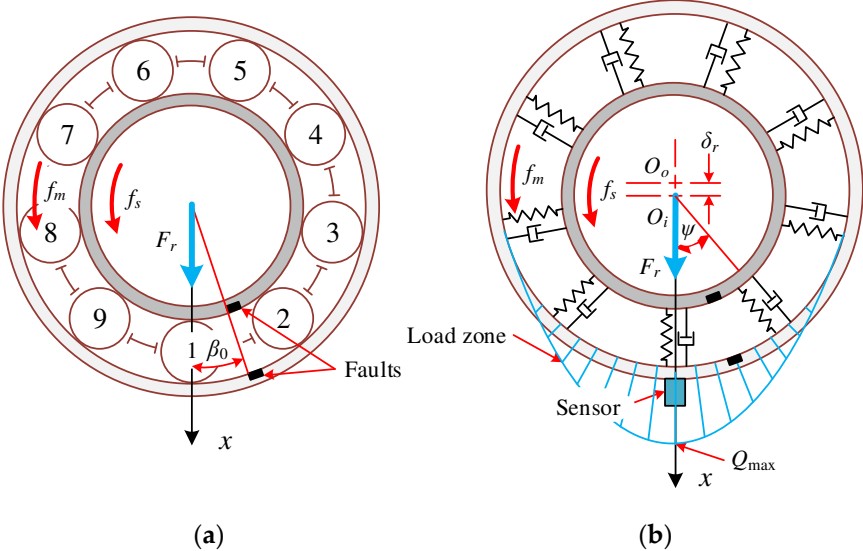

(a)                                  (b)

**Figure 1.** Schematic diagram of bearings under radial load. (**a**) Schematic diagram of the structure; (**b**) Simplified spring-damper model.

The situation when the rolling element falls into the fault position is shown in Figure 2.

In Figure 2, $h_s$ is the maximum theoretical distance that the rolling element can fall when it falls into the fault position, $r$ is the radius of the rolling element, $r_o$ is the radius of the outer raceway, $r_i$ is the radius of the inner raceway, $b$ is the fault width, and $h_d$ is the fault depth.

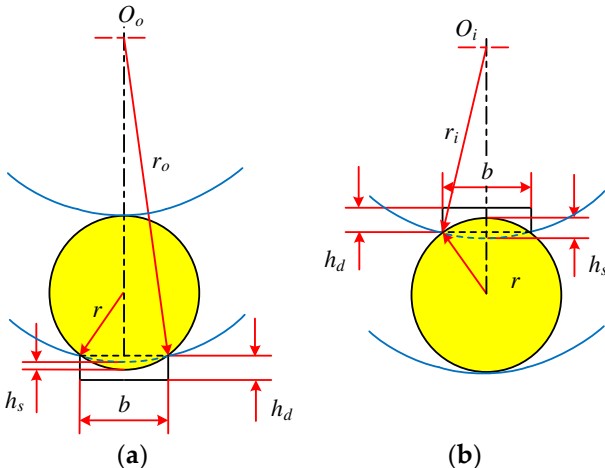

**Figure 2.** Structural diagram when the rolling element falls into the fault position. (**a**) Outer race fault; (**b**) inner race fault.

In the case of $b < r$, the maximum theoretical geometric distance of the collapse when the rolling element falls into the fault position is

$$h_s = \begin{cases} h_d & h_s \geq h_d & \text{Bottoming out} \\ C_d + C_i & \text{Inner race} \quad h_s < h_d \\ C_d - C_o & \text{Outer race} \quad h_s < h_d \end{cases} \quad \text{Not bottoming out} \tag{1}$$

In the formula,

$$C_d = r - \sqrt{r2 - \left(\frac{b}{2}\right)^2} \tag{2}$$

$$C_{i,o} = r_{i,o} - \sqrt{r_{i,o}^2 - \left(\frac{b}{2}\right)^2} \tag{3}$$

where subscripts $i$, $o$ correspond to the inner and outer races, respectively.

The central angle corresponding to the fault width is

$$\Delta\beta = \begin{cases} 2\arcsin\frac{b}{2r_i} & \text{Inner race} \\ 2\arcsin\frac{b}{2r_o} & \text{Outer race} \end{cases} \tag{4}$$

According to the deformation coordination conditions, the radial contact deformation at any azimuth $\psi$ is given by [31]

$$\delta_\psi = \delta_{\max}\left[1 - \frac{1}{2\varepsilon}(1 - \cos\psi)\right] \tag{5}$$

where $\delta_{\max}$ is the maximum value of contact deformation at $\psi = 0°$ and $\varepsilon$ is the load distribution range coefficient [15,23].

$$\varepsilon = \frac{1}{2}\left(1 - \frac{P_d}{2\delta_r}\right) \tag{6}$$

where $P_d$ is the radial clearance and $\delta_r$ is the maximum relative radial displacement of the inner and outer races at $\psi = 0°$ [15,31].

$$\delta_r = \frac{P_d}{2} + \delta_{\max} \tag{7}$$

Under the action of radial load $F_r$, the angular range of the load zone, is [24,31]

$$\psi_L = \pm \arccos\left(\frac{P_d}{2\delta_r}\right) \tag{8}$$

The relationship between the contact deformation $\delta_\psi$ and the contact force $Q(\psi)$ is [14–16,28]

$$Q(\psi) = K\delta_\psi^{1.5} \tag{9}$$

where $K$ is the radial contact stiffness coefficient at a single rolling element [21,22].

$$K = \left[\frac{1}{\left(1/K_{\rho i}\right)^{\frac{2}{3}} + \left(1/K_{\rho o}\right)^{\frac{2}{3}}}\right]^{1.5} \tag{10}$$

where $K_{\rho i}$ and $K_{\rho o}$ are the radial contact stiffness coefficients of the rolling elements and the inner and outer raceways, please refer to the literature [31] for specific calculations.

Therefore, the contact force $Q(\psi)$ can be expressed as [15,23]

$$Q(\psi) = Q_{\max}\left[1 - \frac{1}{2\varepsilon}(1 - \cos\psi)\right]^{1.5} \tag{11}$$

For ball bearings subjected to a single radial load [31]

$$Q_{\max} = \begin{cases} \frac{4.37 F_r}{Z\cos\alpha} & P_d = 0 \\ \frac{5 F_r}{Z\cos\alpha} & P_d \neq 0 \end{cases} \tag{12}$$

where $Z$ is the number of rolling elements and $\alpha$ is the contact angle.

When the rolling element with an angle of $\psi$ falls into the fault position, according to the geometric relationship between the amount of the contact deformation $\delta_\psi$ and the fall distance $h_s$, the retention factor of contact deformation is

$$\gamma = \left(1 - \frac{h_s}{\delta_\psi}\right)^{1.5} \tag{13}$$

(1)　At $h_s < \delta_\psi$, then $0 < \gamma < 1$, that is, the contact deformation at the rolling element is partially released, providing a partially effective contact load.

(2)　At $\delta_\psi \leq h_s$, then $\gamma = 0$, that is, the contact deformation at the rolling element is all released, and the effective contact load cannot be provided.

The change of retention factor $\gamma$ with distance $h_s$ is shown in Figure 3.

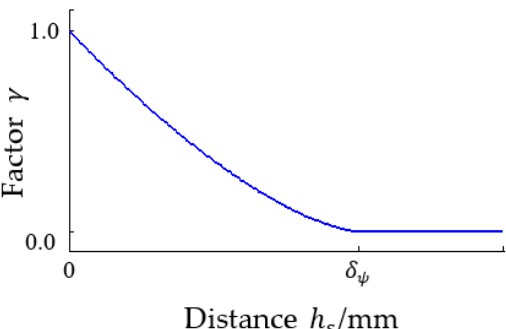

**Figure 3.** Change of contact deformation retention factor $\gamma$.

Figure 3 shows that with the increase in $h_s$, the release of the deformation increases gradually, and the retention factor $\gamma$ decreases gradually. When $h_s \geq \delta_\psi$, the amount of

all deformation is released, the rolling elements cannot provide effective support, and the retention factor $\gamma$ is reduced to 0.

From Equation (9), it can be seen that the contact load when a rolling element with an angle of $\psi$ falls into the fault position is

$$Q(\psi) = K\gamma\delta_\psi^{1.5} \tag{14}$$

Let the No.1 rolling element be located in this position when $\psi = 0°$, as shown in Figure 4a. The angle at which the $i$-th rolling element rotates with the cage is [20,21]

$$\theta_{mi} = \theta_m + (i-1)\frac{2\pi}{Z} \tag{15}$$

where the cage's rotation angle is $\theta_m = 2\pi f_m t$.

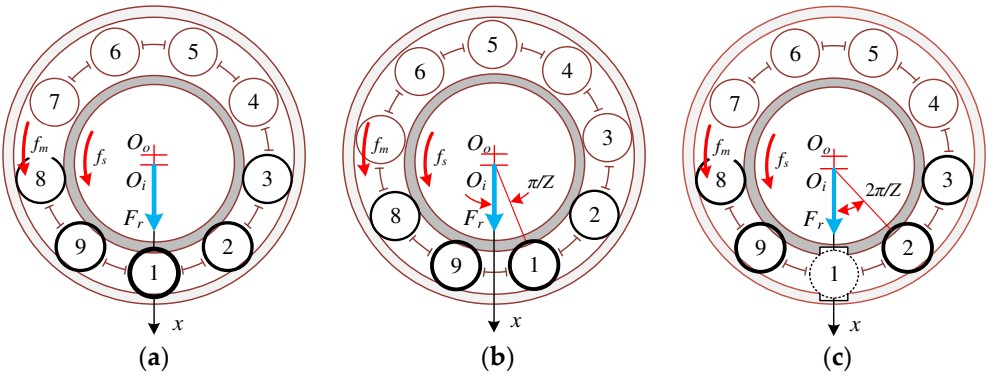

(a)    (b)    (c)

**Figure 4.** Schematic diagram of the change of rolling element support. (**a**) Normal odd number support; (**b**) normal even number support; (**c**) support in case of fault.

Cause

$$\lambda = \begin{cases} 1 & \mathrm{mod}(\theta_{mi}, 2\pi) \in [-\beta_L, \beta_L] \\ 0 & \text{other} \end{cases} \tag{16}$$

where $\mathrm{mod}(\cdot)$ is the remainder operation; parameter $\lambda$ determines whether the rolling element is located in the load zone.

When the outer race fails, the contact stiffness at the $i$-th rolling element is

$$k = \lambda K \begin{cases} \gamma & \mathrm{mod}(\theta_{mi}, 2\pi) \in [\beta_0, \beta_0 + \Delta\beta] \\ 1 & \text{other} \end{cases} \tag{17}$$

When the inner race fails, the rotation angle of the fault point with the inner race is [23]

$$\theta_s = 2\pi f_s t + \beta_0 \tag{18}$$

The contact stiffness at the $i$-th rolling element is

$$k = \lambda K \begin{cases} \gamma & \mathrm{mod}(\theta_{mi}, 2\pi) \in [\mathrm{mod}(\theta_s, 2\pi), \ \mathrm{mod}(\theta_s, 2\pi) + \Delta\beta] \\ 1 & \text{other} \end{cases} \tag{19}$$

To satisfy the static equilibrium relationship between the system and the radial load $F_r$, $F_r$ must be equal to the sum of the components of loads of each rolling element [15].

$$F_r = \sum_{i=1}^{Z} Q(\psi_i) \cos\psi_i \tag{20}$$

where $\psi_i$ is the azimuth of the $i$-th rolling element.

The static equilibrium relationship of the system when there is no fault can be expressed as

$$F_r = k_{eqa}\delta_{\max}^{1.5} \tag{21}$$

where $k_{eqa}$ is the total effective stiffness in the x-direction when there is no fault.

$$k_{eqa} = \lambda K \sum_{i=1}^{Z} \left[ 1 - \frac{1}{2\varepsilon}(1 - \cos\psi_i) \right]^{1.5} \cos\psi_i \tag{22}$$

The effective contact stiffness of the *i*-th rolling element in the x-direction can be expressed as

$$k_i = \lambda K \left[ 1 - \frac{1}{2\varepsilon}(1 - \cos\psi_i) \right]^{1.5} \cos\psi_i \tag{23}$$

In particular, when the radial clearance $P_d$ is 0 and a rolling element is located directly below the radial load $F_r$, there is just an odd number of rolling elements to support, as shown in Figure 4a, and the total effective stiffness of the system is

$$k_{eqa} = K \left( 1 + 2 \sum_{\psi_i = 2\pi/Z}^{\pi/2} \cos^{2.5}\psi_i \right) \tag{24}$$

When the *j*-th rolling element falls into the outer or inner race fault position, the static equilibrium relationship of the system is

$$F_r = k_{eqb}\left(\delta_{\max}'\right)^{1.5} \tag{25}$$

where $\delta_{\max}'$ is the nominal maximum contact deformation in the load zone at $\psi = 0°$; $k_{eqb}$ is the total effective stiffness in the x-direction in case of fault.

$$k_{eqb} = \lambda K \left\{ \sum_{\substack{i=1 \\ i \neq j}}^{Z} \left[ 1 - \frac{1}{2\varepsilon}(1 - \cos\psi_i) \right]^{1.5} \cos\psi_i + \gamma \left[ 1 - \frac{1}{2\varepsilon}(1 - \cos\psi_j) \right]^{1.5} \cos\psi_j \right\} \tag{26}$$

The effective contact stiffness of the *j*-th rolling element in the x-direction can be expressed as

$$k_j = \lambda \gamma K \gamma \left[ 1 - \frac{1}{2\varepsilon}(1 - \cos\psi_j) \right]^{1.5} \cos\psi_j \tag{27}$$

In particular, when the radial clearance $P_d$ is 0, the *j*-th rolling element and point of the outer or inner race fault are directly below the radial load $F_r(\psi_j = 0°)$, and the total effective stiffness of the system is

$$k_{eqb} = K \left( \sum_{\psi_i = 2\pi/Z}^{\pi/2} 2\cos^{2.5}\psi_i + \gamma \right) \tag{28}$$

Further, if $\delta_j \leq h_s$, then an even number of rolling elements are supported, as shown in Figure 4c, and the total effective stiffness of the system is

$$k_{eqb} = 2K \sum_{\psi_i = 2\pi/Z}^{\pi/2} \cos^{2.5}\psi_i \tag{29}$$

Therefore, the total effective stiffness of the system is

$$k_{eq} = \begin{cases} k_{eqa} & \text{Fault} - \text{free} \\ k_{eqb} & \text{Fault} \end{cases} \tag{30}$$

Take a deep groove ball bearing with nine rolling elements in Figure 4 as an example to illustrate the difference between Equations (24) and (29). Figure 4a shows that the normal bearing with exactly one rolling element located directly below the radial load $F_r$ and the number of load-carrying rolling elements comprises odd numbers. The rolling elements are rotated with the cage to the position of Figure 4b, and the number of load-carrying rolling elements is an even number; the azimuth of the No. 9 and No. 1 rolling elements on both sides of the radial load $F_r$ are $+\pi/Z$ and $-\pi/Z$, respectively.

In Figure 4c, the No. 1 rolling element, located below the radial load $F_r$, falls into the inner or outer race fault position and $\delta'_{\max} < h_s$. When the contact deformation is fully released, the number of load-carrying rolling elements suddenly changes from an odd number to an even number, and the azimuth of the No.9 and No.2 rolling elements on both sides of the radial load $F_r$ are $+2\pi/Z$ and $-2\pi/Z$, respectively. In this case, the lack of support for the No. 1 rolling element that provides the largest radial stiffness causes the total effective stiffness to decrease greatly so that the contact deformation of other load-carrying rolling elements increases to rebalance the radial load $F_r$.

The system is simplified in the $x$-direction to a single-degree-of-freedom system with variable stiffness under radial load $F_r$, as shown in Figure 5.

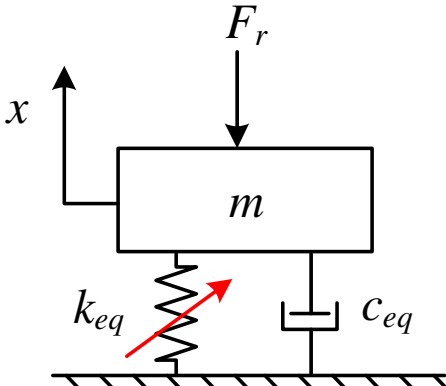

**Figure 5.** Simplified vibration model of bearing system.

In Figure 5, $k_{eq}$ is the coefficient of the total effective stiffness, the slanted arrow indicates that the value is variable, $c_{eq}$ is the damping coefficient [27,32], and $m$ is the mass of the outer or inner race.

In the case where there is no force on the bearing, the system coordinate system is established with the variable $x$ as the origin when the inner and outer races are concentric.

The differential equation of motion for the system is

$$m\ddot{x} + c_{eq}\dot{x} + k_{eq}x = F_r \tag{31}$$

Combining the above formulas, the differential Equation (31) is solved using the fourth-order Runge–Kuta method to obtain the vibration response of the system.

### 3. Simulation and Experimentation

*3.1. Bearing Parameters and Conditions*

Taking the deep groove ball bearing of SKF6205 as an example, the bearing parameters are shown in Table 1.

**Table 1.** SKF6205 Bearing parameters.

| Parameter | Value |
|---|---|
| number of rolling elements $Z$ | 9 |
| diameter of rolling element $d$/mm | 7.94 |
| pitch diameter $D$/mm | 39 |
| contact angle $\alpha/^\circ$ | 0 |
| radius of inner raceway $r_i$/mm | 15.53 |
| radius of outer raceway $r_o$/mm | 23.47 |
| radius of inner groove $R_i$/mm | 4.089 |
| radius of outer groove $R_o$/mm | 4.169 |
| damping coefficient $c_{eq}$/Ns/m | 200 |
| mass $m$/kg | 0.3 |
| radial load $F_r$/N | 490.4 |

The initial condition of the rolling bearing is shown in Figure 4c. At this time, the inner race rotates counterclockwise, and the initial angle of the inner and outer race faults $\beta_0$ is $0^\circ$. The No. 1 rolling element is located below the radial load $F_r$, and $\delta'_{\max} < h_s$, the radial clearance $P_d$ is 0, and the angle of the load zone $\psi_L$ is $\pm 90^\circ$.

### 3.2. Simulation of Total Effective Stiffness $k_{eq}$

From the simulation of Equations (22) and (30), the simulation of the change of the total effective stiffness of the system with the angle of rotation of the cage $\theta_m$ during fault-free, inner race, and outer race faults is obtained as shown in Figure 6. The numbers in Figure 6b,c indicate the serial number of the rolling elements through the fault location.

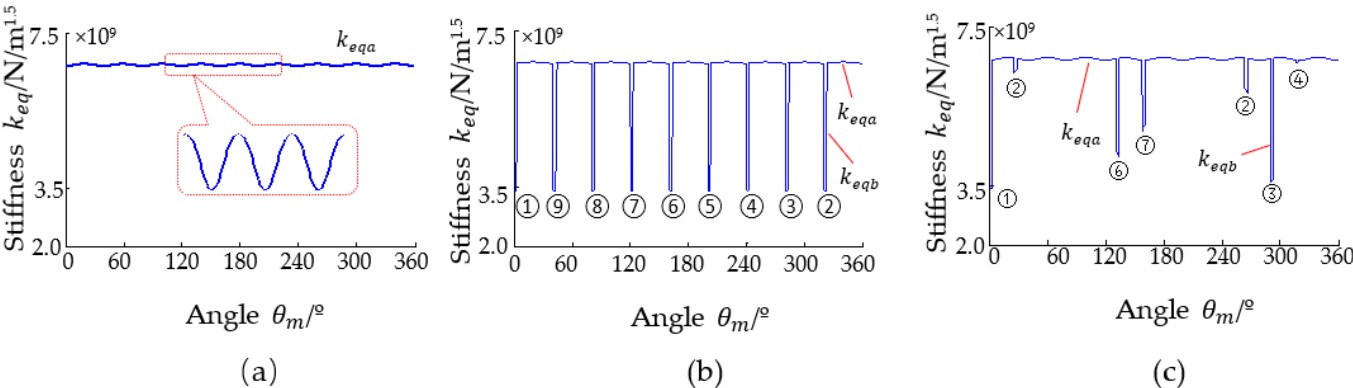

**Figure 6.** Simulation of the change in total effective stiffness of the system. (**a**) Fault-free; (**b**) outer race fault; (**c**) inner race fault.

As shown in Figure 6a, when there is no fault, each rolling element rotates with the cage, so that the number of carrying rolling elements in the x-direction changes periodically from odd numbers (Figure 4a) to even numbers (Figure 4b), and then to odd numbers (Figure 4a). Additionally, the total effective stiffness of the system $k_{eq}$ fluctuates periodically within a small range.

Figure 6b shows the change in the total effective stiffness $k_{eq}$ of the system when the outer race has fault. The outer race fault is located in the load zone and the relative radial load $F_r$, and the position of the load zone is fixed. When each rolling element falls into the fault position in turn, the overall support situation of other rolling elements is the same. As shown in Figure 7, the total effective stiffness is reduced from $k_{eqa}$ to $k_{eqb}$, and the $k_{eqb}$ is equal in amplitude. The value of $k_{eqb}$ is related to the location angle of the fault $\beta_0$, independent of the angle of rotation of the inner race.

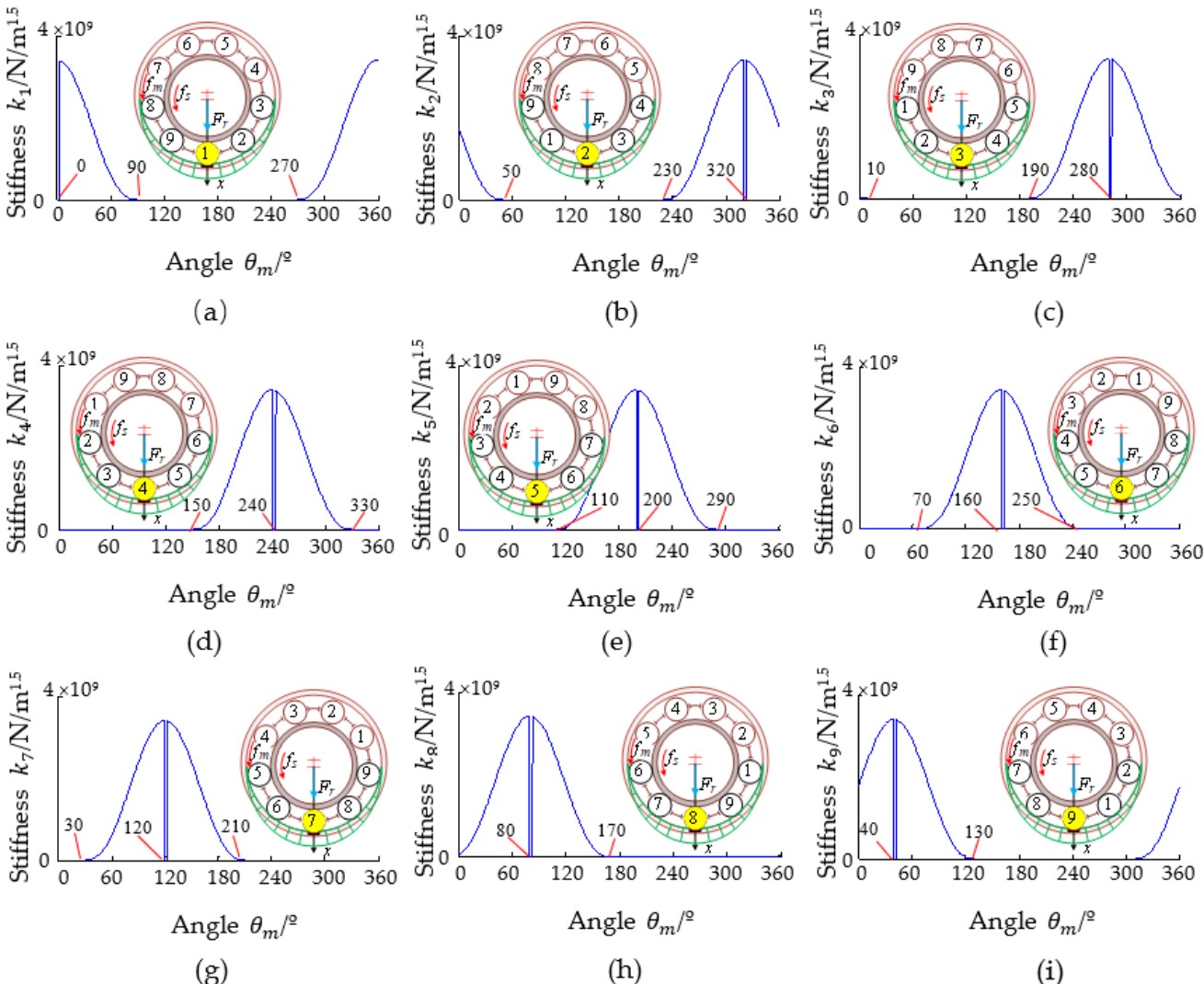

**Figure 7.** Contact stiffness of each rolling element in case of the outer race fault. (**a**) No. 1 rolling element; (**b**) No. 2 rolling element; (**c**) No. 3 rolling element; (**d**) No. 4 rolling element; (**e**) No. 5 rolling element; (**f**) No. 6 rolling element; (**g**) No. 7 rolling element; (**h**) No. 8 rolling element; (**i**) No. 9 rolling element.

Figure 6b shows the change in the total effective stiffness $k_{eq}$ of the system when the inner race has fault. Because the fault rotates with the inner race, its position varies periodically with respect to the radial load $F_r$ and the position of the load zone. When the rolling element and the fault point are in the load zone and meet, it causes the contact stiffness at the rolling element to change, resulting in the total effective stiffness decreasing from $k_{eqa}$ to $k_{eqb}$. The value of $k_{eqb}$ is related to the position of the rolling element when it meets the fault point in the load zone. The $k_{eqb}$ is variable in amplitude and it is modulated by the rotation of the inner race.

*3.3. Simulation of Stiffness, Contact Force, and Contact Deformation of Each Rolling Element in Case of Outer Race Fault*

From the simulation of Equations (23) and (27), it is obtained that the change of effective contact stiffness of each rolling element in the x-direction during the cage rotates one turn in case of the outer race fault as shown in Figure 7. Each rolling element calculates its angle of rotation with the cage starting from the initial azimuth. The key position of each rolling element entering and exiting the load zone and falling into the fault position is shown in Table 2.

**Table 2.** The state of each rolling element rotates one week with the cage in case of the outer race fault.

| Rolling Element | 1 | 2 | 3 | 4 | 5 | 6 | 7 | 8 | 9 |
|---|---|---|---|---|---|---|---|---|---|
| Initial azimuth $\psi_{0i}/°$ | 0 | 40 | 80 | 120 | 160 | 200 | 240 | 280 | 320 |
| Whether it is located in the load zone | Yes | Yes | Yes | No | No | No | No | No | No |
| Rotation angle required to enter the load zone $\Delta\theta_{mi\_in}/°$ | | | | 150 | 110 | 70 | 30 | | |
| Rotation angle required to exit load zone $\Delta\theta_{mi\_out}/°$ | 90 | 50 | 10 | 330 | 290 | 250 | 210 | 170 | 130 |
| The rotation angle of the cage where the rolling element meets the fault $\Delta\theta_{mi}/°$ | 0 | 320 | 280 | 240 | 200 | 160 | 120 | 80 | 40 |

Figure 7a shows that the No. 1 rolling element falls into the fault position and is unable to provide support at this time. After the No. 1 rolling element is out of the fault with the rotation of the cage, its azimuth gradually increases, making the effective contact stiffness provided gradually decrease. The No. 1 rolling element rotates 90° with the cage and then exits the load zone, and the effective contact stiffness drops to 0 at this time. However, the initial azimuth of the No. 9 rolling element is 320°; its azimuth gradually decreases with the rotation of the cage, making the effective contact stiffness provided gradually increase. The No. 9 rolling element falls into the fault position when it rotates 40° with the cage, the effective contact stiffness drops to 0 at this time. After the No. 9 rolling element becomes out of the fault with the rotation of the cage, its azimuth gradually increases, making the effective contact stiffness provided gradually decrease. The No. 9 rolling element rotates 90° with the cage and then exits the load zone; the effective contact stiffness drops to 0 at this time, as shown in Figure 7i. The case of other rolling elements is similar and will not be repeated.

Each rolling element is 40° apart, and they fall into the fault position in turn when the cage rotates counterclockwise. Therefore, the entire rolling bearing in the load zone is four rolling elements to provide support, and the support situation is identical; only the rolling element serial number is different, and the total effective stiffness $k_{eqb}$ is equal in this case. By superimposing the effective contact stiffness of all rolling elements in the *x*-direction, the total effective stiffness is obtained as shown in Figure 6b.

Taking the case of the No. 5 rolling element falling into the fault position in Figure 7e as an example, analyze the changes of the contact deformation and the contact force of the No. 3, No. 4, No. 6, and No. 7 load-carrying rolling elements in the load zone under fault-free and fault conditions. From the simulation of Equations (5), (13), and (16), the contact force and the contact deformation are obtained as shown in Figure 8. From the simulation of Equations (9), (23), and (27), the contact force and the contact deformation are obtained as shown in Figure 9.

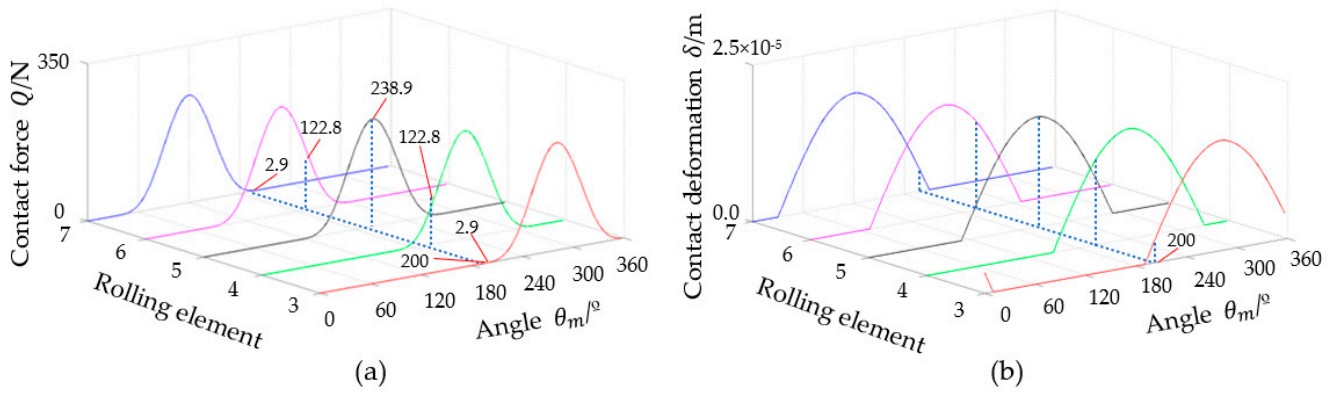

**Figure 8.** Contact force and contact deformation of No. 3, No. 4, No. 5, No. 6, and No. 7 rolling elements in case of fault–free. (**a**) Contact force; (**b**) contact deformation.

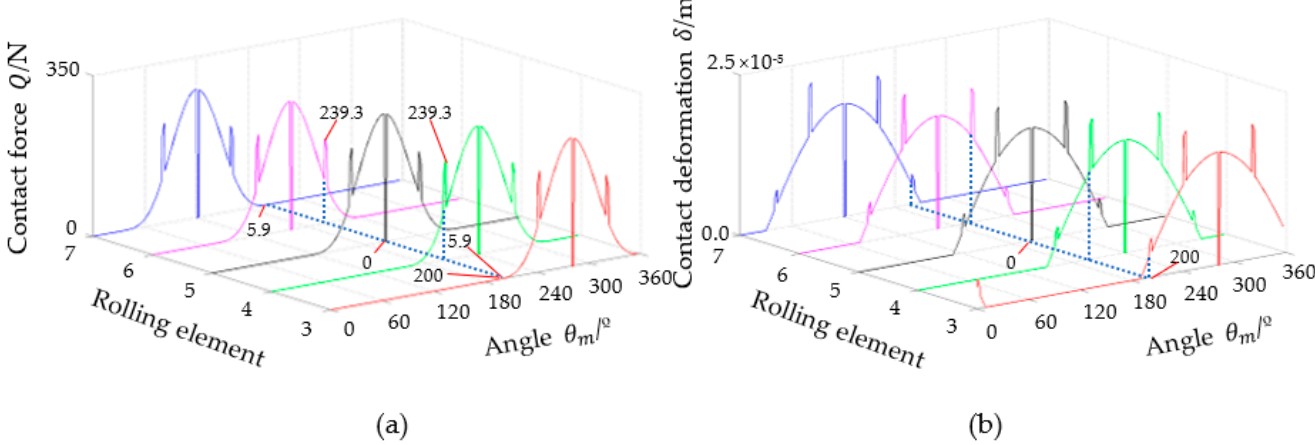

**Figure 9.** Contact force and contact deformation of No. 3, No. 4, No. 5, No. 6, and No. 7 rolling elements in case of outer race fault. (**a**) Contact force; (**b**) contact deformation.

As can be seen from Figure 8, the No. 5 rolling element is located below the radial load $F_r$ after rotating 200° with the cage, and it is subjected to the largest load and contact deformation in the load zone; the No. 4 and No. 6 rolling elements are next, and the No. 3 and No. 7 rolling elements are the smallest.

As can be seen from Figure 9, when the No. 5 rolling element falls into the fault position, its contact deformation $\delta_5$ is all released, and the contact deformation $\delta_5$ and contact force $Q_5$ are 0. To rebalance the external radial load $F_r$, the No. 4 and No. 6 rolling elements, which are closest to the No. 5 rolling element, become the most important load-carrying rolling elements. Additionally, their contact force and contact deformation increase greatly, and because the angle of the No. 4 rolling element and the radial load $F_r$ and the angle of the No. 6 rolling element and the radial load $F_r$ are equal, their increase is the same. The contact force and contact deformation of the No. 3 and No. 7 rolling elements, which are far from the No. 5 rolling element, also have an equal increase, but because the angle of the No. 3 rolling element and $F_r$ and the angle of the No. 7 rolling element and $F_r$ are large, their increase is smaller.

*3.4. Simulation of the Stiffness, Contact Force, and Contact Deformation of Each Rolling Element in Case of the Inner Race Fault*

From the simulation of Equations (23) and (27), the change of stiffness of each rolling element as it rotates with the cage and meets the inner race fault is obtained as shown in Figure 10, the key position of each rolling element as it meets the fault is shown in Table 3, and the angle of entering and exiting the load zone is the same as Figure 7 and Table 2.

**Table 3.** The state of each rolling element when rotating with the cage in case of inner race fault.

| Rolling Element | 1 | 2 | 3 | 4 | 5 | 6 | 7 | 8 | 9 |
|---|---|---|---|---|---|---|---|---|---|
| Initial azimuth $\psi_{0i}/°$ | 0 | 40 | 80 | 120 | 160 | 200 | 240 | 280 | 320 |
| Whether it is located in the load zone | Yes | Yes | Yes | No | No | No | No | Yes | Yes |
| Rotation angle of the cage where the rolling element meets the fault $\Delta\theta_{mi}/°$ | 0 | 25.2 | 51.7 | 78.2 | 104.7 | 131.1 | 157.6 | 184.1 | 210.5 |
| Rotation angle of the fault when the rolling element meets the fault $\theta_{si}/°$ | 0 | 65.2 | 131.7 | 198.2 | 264.7 | 331.1 | 397.6 | 494.1 | 530.5 |
| Whether it is located in the load zone when the rolling element meets the fault | Yes | Yes | No | No | No | Yes | Yes | No | No |

As can be seen from Figure 10, the change in stiffness in the case of the inner race fault is much more complex than in the case of the outer race fault, and the position of each rolling element that falls into the fault position is related to the inner race rotation. Each rolling element meets the inner race fault every 26.5° as it rotates with the cage, the inner race rotation angle is 66.5° at this time, and the number of rolling elements in the load zone is 4 or 5. If the inner race fault is rotated outside the load zone, although it meets the rolling

element, they are not in contact, so there is no change in the effective contact stiffness. Like the No. 3, No. 4, No. 5, No. 8, and No. 9 rolling elements in Figure 10, the contact force and contact deformation are 0 at this time.

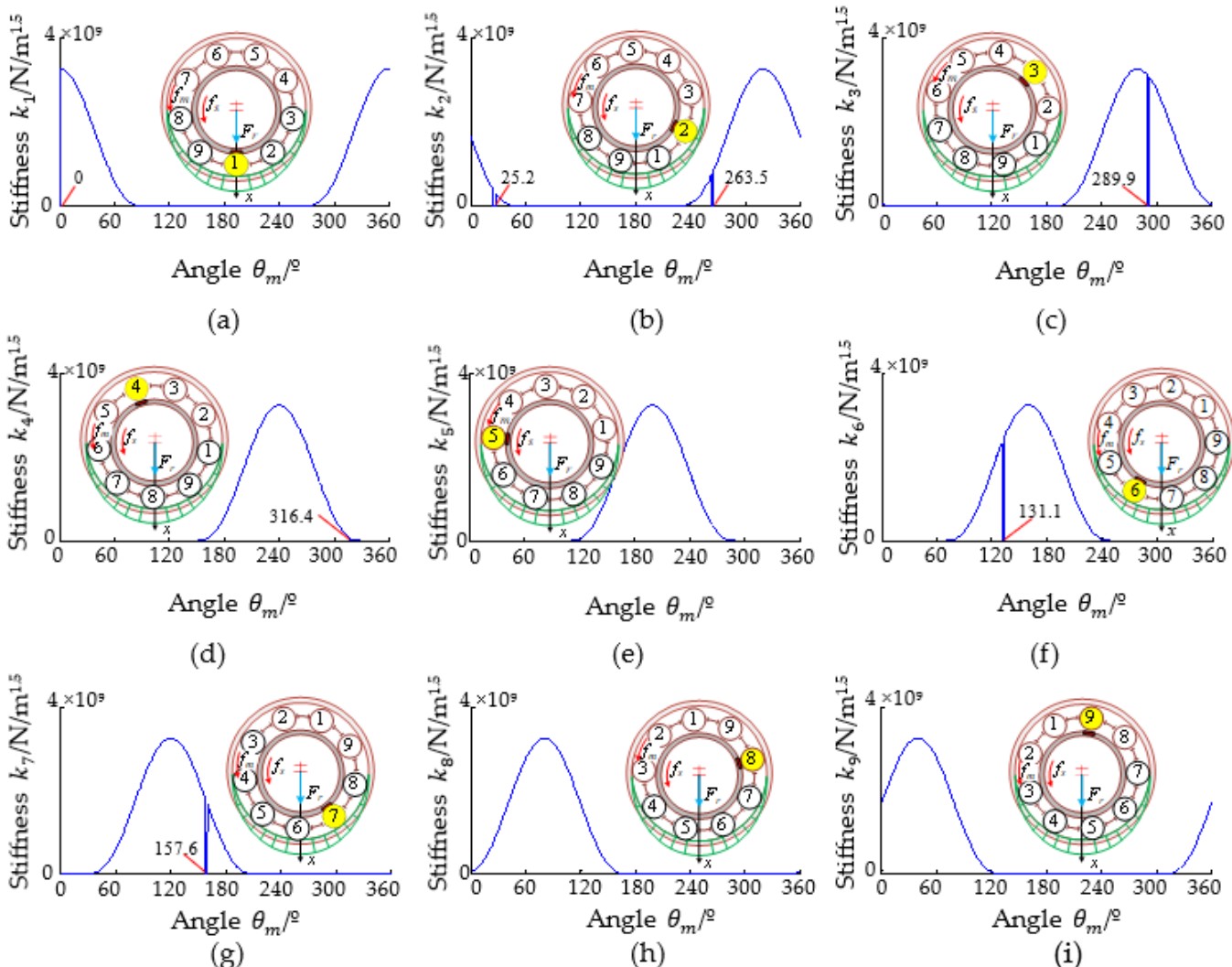

**Figure 10.** Contact stiffness of each rolling element in case of inner race fault. (**a**) No. 1 rolling element; (**b**) No. 2 rolling element; (**c**) No. 3 rolling element; (**d**) No.4 rolling element; (**e**) No. 5 rolling element; (**f**) No. 6 rolling element; (**g**) No. 7 rolling element; (**h**) No.8 rolling element; (**i**) No. 9 rolling element.

From the simulation of Equations (5), (13), and (16), the change in the contact deformation of each rolling element in the case of the inner race fault is obtained as shown in Figure 11. From the simulation of Equations (9), (23) and (27), the change in contact force is obtained as shown in Figure 12. As can be seen in Figures 11 and 12, due to the fault rotating with the inner race, the position of each rolling element when it falls into the fault position is different, and the contact deformation and contact force increase in other rolling elements in the load zone are also different.

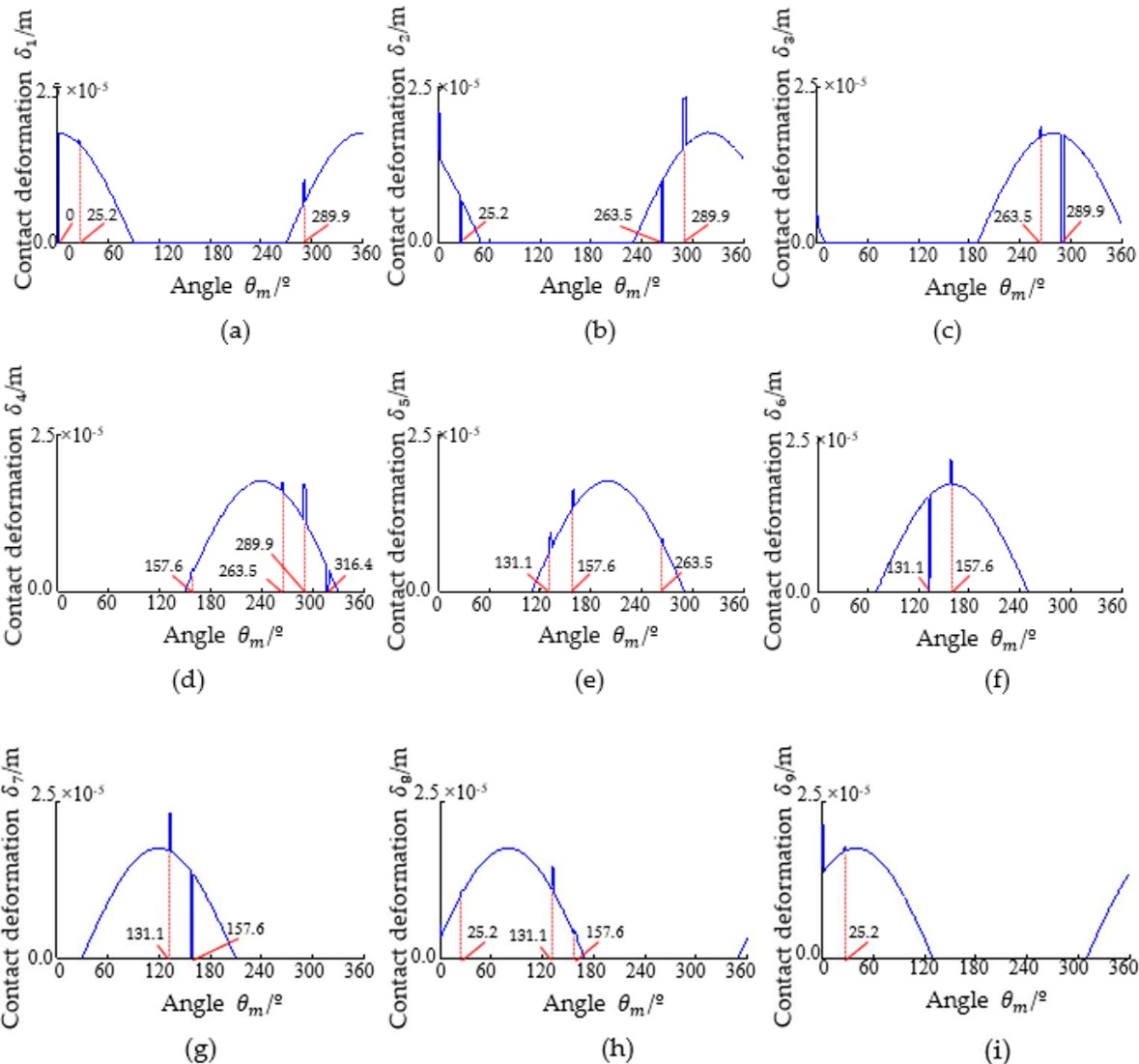

**Figure 11.** Change of contact deformation of each rolling element in case of the inner race fault. (**a**) No. 1 rolling element; (**b**) No. 2 rolling element; (**c**) No. 3 rolling element; (**d**) No. 4 rolling element; (**e**) No. 5 rolling element; (**f**) No. 6 rolling element; (**g**) No. 7 rolling element; (**h**) No. 8 rolling element; (**i**) No. 9 rolling element.

Take the No. 6 rolling element falling into the fault position in Figure 10f as an example. From the simulation of Equations (9), (23), and (27), the changes in contact deformation and contact force of the No. 5, No. 7, and No. 8 rolling elements in the load zone at this time as shown in Figure 13. When the No. 6 rolling element falls into the inner race fault position after rotating 131.1° with the cage, its contact deformation $\delta_6$ is all released, and its contact deformation $\delta_6$ and contact force $Q_6$ are 0. To balance the external radial load $F_r$, the angle between the No. 7 rolling element and the radial load $F_r$ is minimal; that is, near the center of the load area, its contact force and contact deformation increase greatly and become the most important load-carrying rolling elements. At the same time, the No. 5 and No. 8 rolling elements are at the edge of the load zone; their contact force and contact deformation also increase, but not much. Similarly, when rolling element No. 7 falls into the inner race fault position after rotating 157.6° with the cage, the rolling elements in the load zone at this time are No. 4, No. 5, No. 6, No. 7, and No. 8, and the changes in the contact deformation and the contact force are similar and will not be repeated.

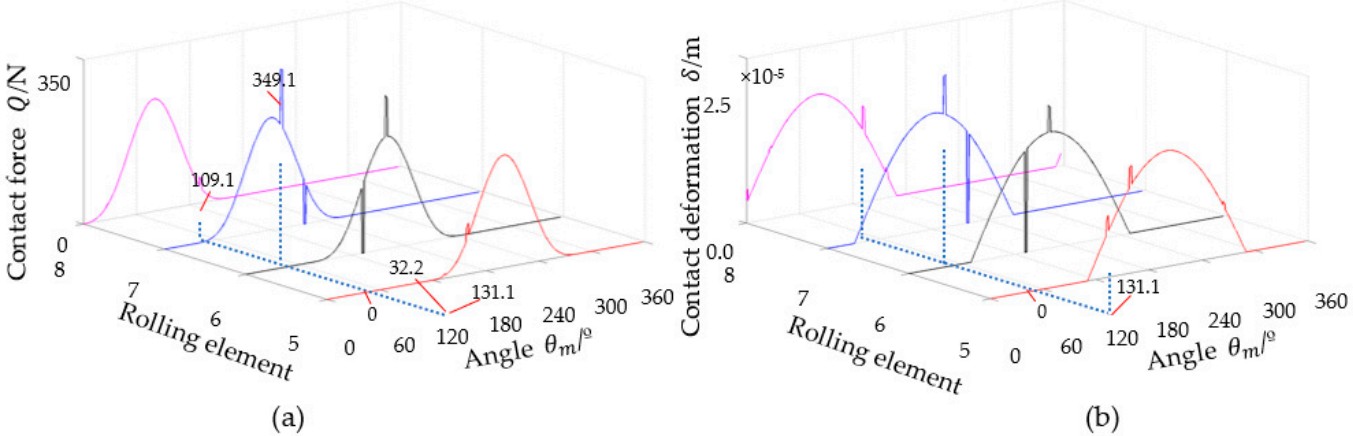

**Figure 12.** Change of the contact force of each rolling element in case of inner race fault. (**a**) No. 1 rolling element; (**b**) No. 2 rolling element; (**c**) No. 3 rolling element; (**d**) No. 4 rolling element; (**e**) No. 5 rolling element; (**f**) No. 6 rolling element; (**g**) No. 7 rolling element; (**h**) No. 8 rolling element; (**i**) No. 9 rolling element.

**Figure 13.** Contact force and contact deformation when the No.6 rolling element falls into the inner race fault position. (**a**) Contact force; (**b**) contact deformation.

*3.5. Simulation and Experiment of System Response*

Fault characteristic frequency of the rolling bearing inner and outer races [11,14]

$$f_i = \frac{Zf_s}{2}\left(1 + \frac{d}{D}\cos\alpha\right) \tag{32}$$

$$f_o = \frac{Zf_s}{2}\left(1 - \frac{d}{D}\cos\alpha\right) \tag{33}$$

where $f_i$ is the fault characteristic frequency of the inner race; $f_o$ is the fault characteristic frequency of the outer race.

The bearing experimental data of the rotor test bench are from Case Western Reserve University (USA). The model of the rolling bearing is SKF6205, and its parameters are shown in Table 1. In the case of the inner race fault, the rotor test stand motor speed is 1721 rpm, namely, 28.68 Hz, and the theoretical calculation of the characteristic frequency $f_i$ of the inner race fault is 155.35 Hz. In the case of the outer race fault, the rotor test stand motor speed is 1725 rpm, namely, 28.75 Hz, and the characteristic frequency $f_o$ is 103.03 Hz. The dimensions of the fault are ∅0.18 mm and 0.28 mm deep, located directly below the bearing. The fault bearing is installed at the drive end, and the sampling frequency is 48 kHz. The manufacturing and installation of the bearing will cause deviations between the measured frequency and the theoretical calculation. In addition, the rotor test bench of our university was used for the experiment, as shown in Figure 14. The model of the rolling bearing is 6000, and its main parameters are shown in Table 4, where the sensor is a KISTLER 8702B25. The motor speed at the time of failure is 900 rpm, i.e., 15.0 Hz, and the theoretical calculation of $f_i$ is 66.55 Hz and $f_o$ is 38.45 Hz. The dimensional width of the failure is 1 mm and 0.7 mm deep, as shown in Figure 15.

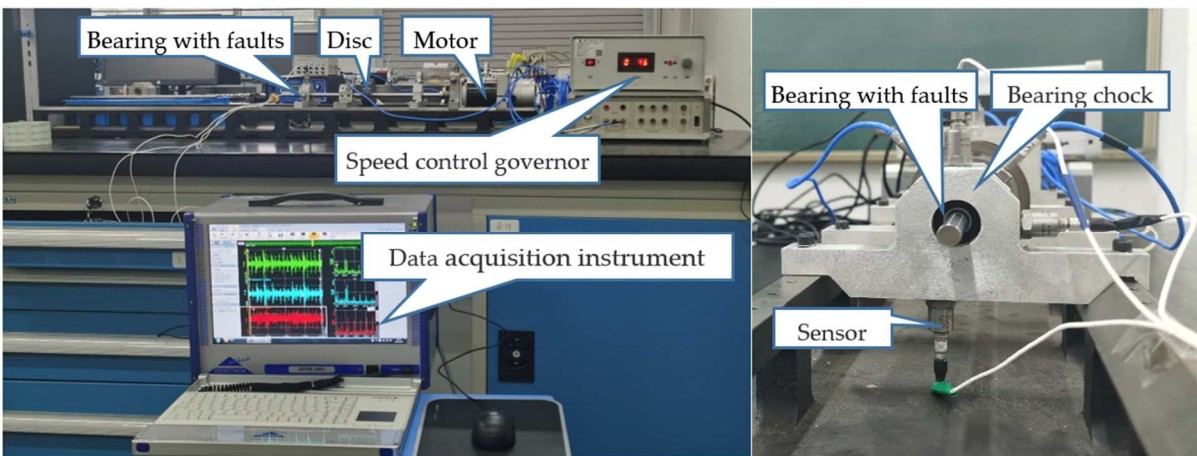

**Figure 14.** Rotor test bench.

**Table 4.** Results of 6000 bearing parameters.

| Parameter | Value |
| --- | --- |
| number of rolling elements Z | 7 |
| diameter of rolling element $d$/mm | 4.762 |
| pitch diameter $D$/mm | 17.8 |
| contact angle $\alpha$/° | 0 |
| radius of inner raceway $r_i$/mm | 6.52 |
| radius of outer raceway $r_o$/mm | 11.28 |
| damping coefficient $c_{eq}$/Ns/m | 100 |
| mass $m$/kg | 0.01 |
| radial load $F_r$/N | 10 |

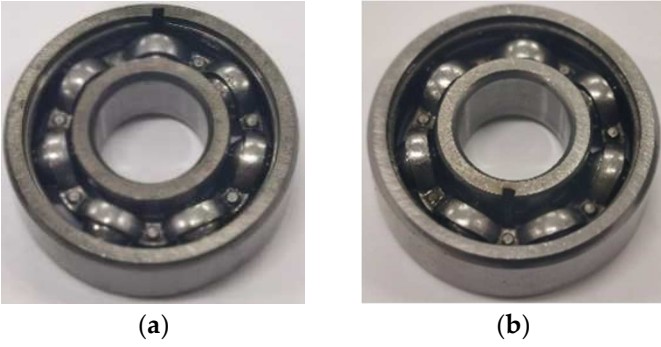

(**a**)　　　　　　　　　　(**b**)

**Figure 15.** Bearing faults. (**a**) Outer race fault; (**b**) inner race fault.

The theoretical simulation uses the variable stiffness model (VEM) of this paper and the more widely used two-degree-of-freedom displacement excitation model (DEM) and compares and analyzes the experimental data. The simulation and experimental results of the local fault of the rolling bearing outer race are shown in Figure 16. The experimental data and the VSM in Figure 16a,c are represented by the left axis, and the DEM is represented by the right axis. In Figure 16b, the experimental data and the VSM and the DEM are represented by the left coordinate axis, and in Figure 16d, the experimental data and the DEM are represented by the left coordinate axis, and the VSM is represented by the right coordinate axis.

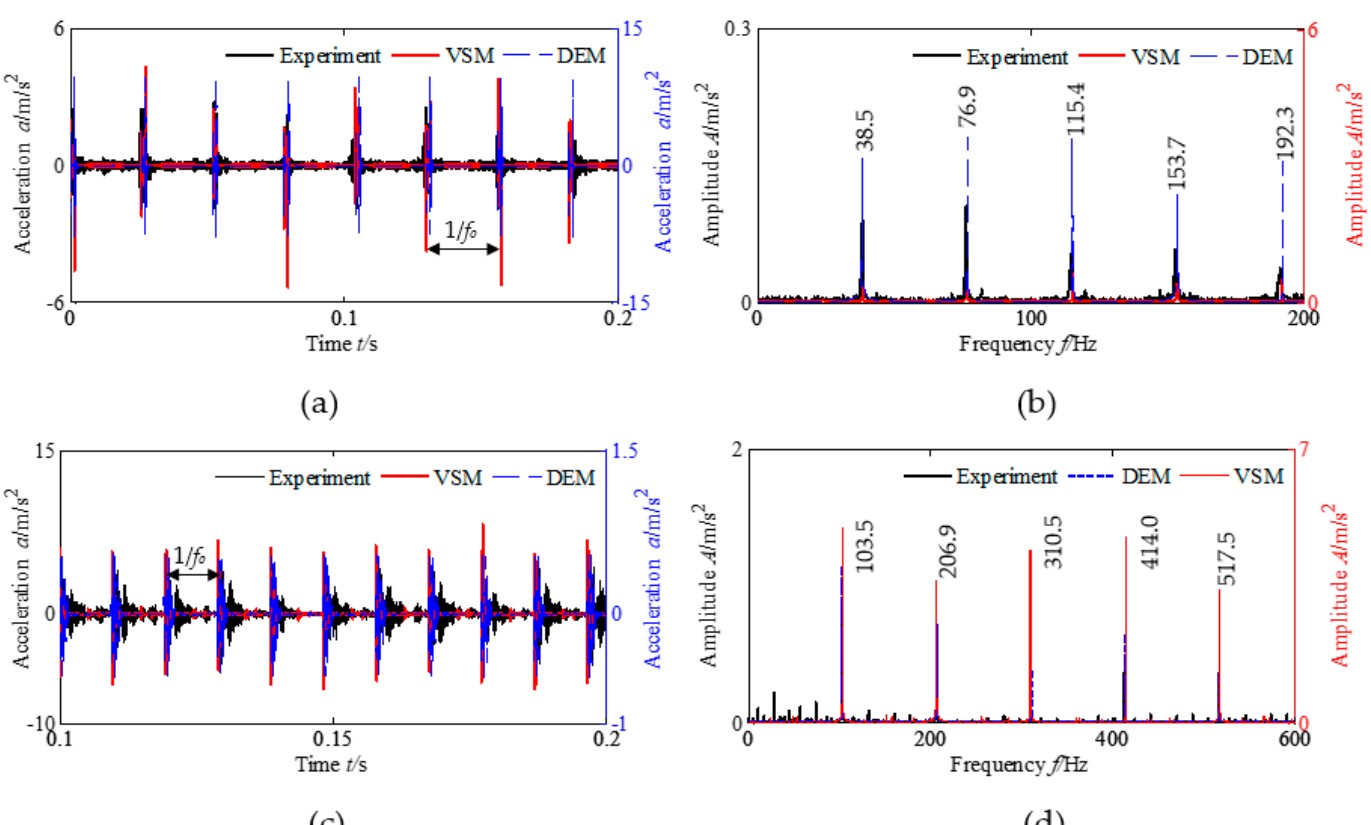

**Figure 16.** Simulation and experiment of the outer race local fault. (**a**) The 6000 time domain waveform; (**b**) 6000 envelope spectrum; (**c**) SKF6205 time domain waveform; (**d**) SKF6205 envelope spectrum.

Figure 16a,c shows the time domain waveform in each impact response position. The VSM and the DEM match with the experimental data. The amplitude of the VSM is basically

the same as the experimental data, but the DEM is more different. In general, the change in each shock response amplitude is small, and the distance between two adjacent amplitudes is $1/f_o$. Because the overall support of the bearing is the same after each rolling element falls into the fault position, the change in total effective stiffness $k_{eq}$ and the change in contact stiffness when each rolling element falls into the fault position are also the same, as shown in Figures 6b and 7, resulting in the same amplitude of the system response under radial load.

Figure 16b,d shows the envelope spectrum. There are significant outer race local fault characteristic frequencies f accompanied by frequency multiplication in the figure. For example, the characteristic frequencies of bearing 6000 are 38.5 Hz, 76.9 Hz, etc., and the characteristic frequencies of bearing SKF6205 are 103.5 Hz, 206.9 Hz, 310.5 Hz, etc. At each fault characteristic frequency position, the VSM and the DEM match with the experimental data, but the amplitude of the DEM in Figure 16b is too large, while the amplitude of the VSM in Figure 16d is slightly larger. Therefore, the results of the VSM are in better agreement with the experimental results and are superior to the DEM.

The simulation and experimental results of the local fault of the rolling bearing inner race are shown in Figure 17. The experimental data in Figure 17 are represented by the left axis, and the VSM and the DEM are represented by the right axis.

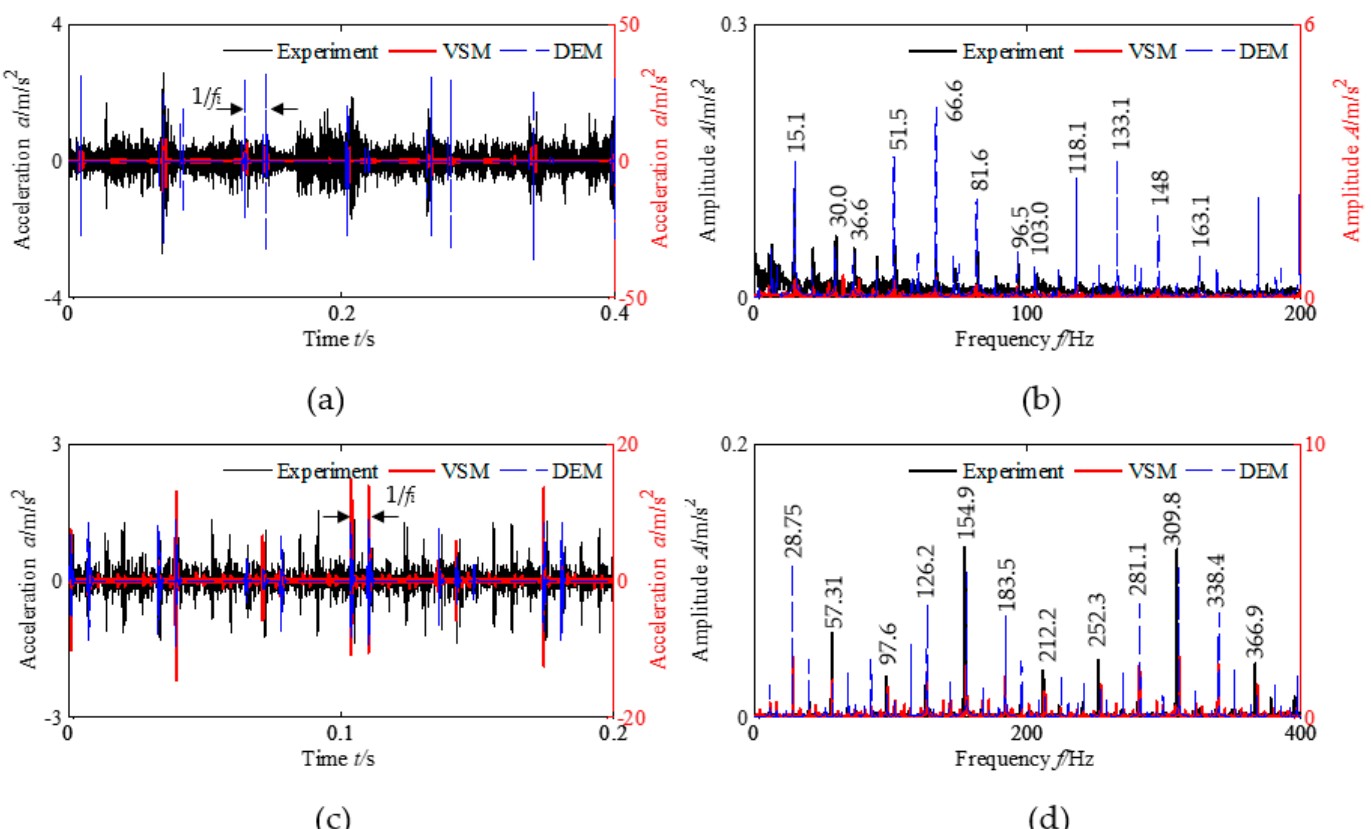

**Figure 17.** Simulation and experiment of the inner race local fault. (**a**) The 6000 time domain waveform; (**b**) 6000 envelope spectrum; (**c**) SKF6205 time domain waveform; (**d**) SKF6205 envelope spectrum.

Figure 17a,c shows the time domain waveform in each impact response position. The VSM and the DEM match the experimental data. The amplitudes of the DEM and the VSM are larger than the experimental data, but the DEM is more different in Figure 17a. In general, each shock response amplitude varies drastically, and the distance between two adjacent amplitudes is $1/f_i$. When each rolling element rotates with the cage and meets the inner race fault, the total effective stiffness $k_{eq}$ is changed, the change in the total effective

stiffness $k_{eq}$ and the change in the contact stiffness when each rolling element falls into the fault position are different, as shown in Figures 6c and 8, and the magnitude of the change of total effective stiffness $k_{eq}$ is modulated by the inner race rotation, resulting in a change in the system response amplitude as well.

Figure 17b,d shows the envelope spectrum. There is a significant inner race rotation frequency $f_s$ accompanied by frequency multiplication as well as fault characteristic frequency $f_i$ and its frequency multiplication in the figure. The side frequency with obvious amplitude is distributed on both sides of the fault characteristic frequency $f_i$ and its frequency multiplication. For example, the characteristic frequencies of bearing 6000 are 66.6 Hz and 133.1 Hz; 36.6 Hz, 51.5 Hz, 81.6 Hz, and 96.5 Hz are the side frequencies of 66.6 Hz; and 15.1 Hz and 30.0 Hz are the inner race rotation frequencies. The characteristic frequencies of bearing SKF6205 are 154.9 Hz and 309.8 Hz; 252.3 Hz, 281.1 Hz, 338.4 Hz, and 366.9 Hz are the side frequencies of 309.8 Hz, and 28.75 Hz and 57.31 Hz are the inner race rotation frequencies. Therefore, the theoretical simulation and experimental results agree that there is a significant modulation of the response amplitude in the case of the inner race fault. At each fault characteristic frequency and its side frequency and rotation frequency positions, the VSM and the DEM match with the experimental data, but the amplitude of the DEM is significantly larger than that of the VSM and experimental data. Therefore, the results of the VSM are in better agreement with the experimental results and are better than the DEM.

The degree of agreement between the simulated and experimental data can be expressed in terms of average coherence:

$$\tau = \frac{1}{N} \sum_{i=1}^{N} \left( \frac{A_i - A_{Ei}}{A_{Ei}} \right) \tag{34}$$

where $N$ is the number of time domain peak samples ($N = 30$), $A_{Ei}$ is the peak of experimental data, and $A_i$ is the peak of the VSM or DEM simulation data.

From Table 5, the amplitude of the VSM is more consistent with the experimental data, and the consistency of the outer ring is the best, with a minimum of 49%; the inner ring is slightly worse, with a minimum of 241%. However, both are significantly better than the DEM, indicating that the VSM is more consistent with the reality.

**Table 5.** Average coherence validation data.

| Position | | $\overline{A_E}$/m/s² | $\overline{A_{VSM}}$/m/s² | $\tau_{VSM}$ | $\overline{A_{DEM}}$/m/s² | $\tau_{DEM}$ |
|---|---|---|---|---|---|---|
| inner race | SKF6205 | 0.93 | 5.31 | 471% | 7.92 | 752% |
| | 6000 | 1.80 | 6.14 | 241% | 26.26 | 1359% |
| outer race | SKF6205 | 3.88 | 6.71 | 73% | 0.54 | −86% |
| | 6000 | 2.58 | 3.85 | 49% | 9.62 | 273% |

In summary, when the rolling elements fall into the fault positions of the inner and outer races, the total effective stiffness of the system will change abruptly. Therefore, to re-balance the external radial load, the contact deformation and contact force of each load-carrying rolling element in the load zone change, and finally, this causes the dynamic system vibration. As the fault position of the outer race is unchanged relative to the radial load and the position of the load zone, the amplitude of the total effective stiffness reduction of the system is alike when each rolling element falls into the fault position with the rotation of the cage. Therefore, the amplitude of the outer loop response is alike, and there is a significant characteristic frequency $f_o$ of outer race faults and its frequency multiplication in the fault characteristic spectrum. However, the position of the inner race fault relative to the radial load and the load zone changes periodically with the rotation of the inner race; the rolling elements in the load zone will change the total effective stiffness of the system due to the inner race fault. Therefore, the total effective support stiffness of the system is affected by the rotation of the inner race, which further leads to the amplitude

of the inner loop response also being affected. In the fault characteristic spectrum of the inner race, there is a significant rotation frequency of the inner race $f_s$ accompanied by frequency multiplication as well as a fault characteristic frequency of the inner race $f_i$ and its frequency multiplication and side frequency.

## 4. Conclusions

By setting up the dynamics modeling of the rolling bearing variable stiffness system with local fault, this paper researches the change in effective contact stiffness, contact deformation, contact forces, and total effective stiffness of the rolling elements. Some conclusions are as follows.

(1) Based on the retention factor of contact deformation, the single-degree-of-freedom variable stiffness model of rolling bearings with fault is proposed, and the dynamics modeling of the variable stiffness of rolling bearings with fault is established.

(2) The contact stiffness of the rolling elements abruptly decreases when the rolling elements fall into the fault position. The contact deformation and contact force of the load-carrying rolling elements in the load zone increases, rebalancing the external radial load while causing a sudden reduction in the total effective stiffness, resulting in the vibration of the system.

(3) When different rolling elements fall into the outer race fault position, the change in the total effective stiffness and the system response are equal in magnitude. Additionally, there is significant outer race fault characteristic frequency and its frequency multiplication in the fault characteristic spectrums. When different rolling elements fall into the inner race fault position, the total effective stiffness is modulated by the inner race rotation and varies dramatically, resulting in the amplitude of the system time domain vibration response also being modulated by the inner race rotation and varying dramatically. Additionally, there are significant inner race rotational frequencies and their frequency multiplications, inner race fault characteristic frequencies and their frequency multiplication, and side frequency in the fault characteristic spectrums.

(4) The VSM is more consistent with the experiment and provides some theoretical basis for the effective diagnosis of rolling bearing faults. The contact deformation retention factor is only for rectangular or circular faults, which has some limitations. In the subsequent research, the VSM will be applied to other types of rolling bearings to expand the application scope of the VSM.

**Author Contributions:** Conceptualization, B.G.; methodology, B.G.; software, W.W.; validation, W.W.; formal analysis, J.Z. (Jianxiao Zheng), Y.H. and J.Z. (Jinhua Zhang); investigation, B.G.; resources, Y.H.; data curation, B.G.; writing—original draft preparation, B.G.; writing—review and editing, B.G. and W.W.; project administration, J.Z. (Jianxiao Zheng); funding acquisition, J.Z. (Jianxiao Zheng), Y.H. and J.Z. (Jinhua Zhang). All authors have read and agreed to the published version of the manuscript.

**Funding:** This work was supported by the Natural Science Basic Research Plan of Shaanxi Province (No. 2023-JC-YB-313); Shaanxi Province Qin Chuangyuan "scientists + engineers" team construction (2022KXJ032); and Shanxi coal group 2022 annual plate level scientific research project (2022SMHKJ-BK-J-14).

**Institutional Review Board Statement:** Not applicable.

**Informed Consent Statement:** Not applicable.

**Data Availability Statement:** Not applicable.

**Conflicts of Interest:** The authors declare no conflict of interest.

## Nomenclature

| | |
|---|---|
| $b$ | fault width (mm) |
| $h_s$ | geometric distance (mm) |
| $h_d$ | fault depth (mm) |
| $d$ | diameter of rolling element (mm) |
| $D$ | pitch diameter of bearing (mm) |
| $\gamma$ | retention factor of contact deformation |
| $f_s, f_m$ | inner race/cage rotation frequency (Hz) |
| $\lambda$ | dimensionless parameters |
| $f_i, f_o$ | fault characteristic frequency of inner/outer races (Hz) |
| $\theta_m$ | rotation angle of the cage (°) |
| $F_r$ | external radial load (N) |
| $\theta_{mi}$ | rotation angle of the $i$-th rolling element with the cage (°) |
| $K, K_{\rho i}, K_{\rho o}$ | contact stiffness coefficient (N/m$^{3/2}$) |
| $\theta_s$ | rotation angle of the fault point with the inner race (°) |
| $k_i, k_j$ | effective contact stiffness (N/m$^{3/2}$) |
| $\delta_{\max}, \delta_\psi, \delta_j$ | maximum/radial contact deformation (mm) |
| $k_{eqa}, k_{eqb}, k_{eq}$ | total effective stiffness (N/m$^{3/2}$) |
| $\delta_r$ | maximum relative radial displacement (mm) |
| $m$ | mass of the outer or inner race (kg) |
| $\varepsilon$ | load distribution range coefficient |
| $r$ | radius of rolling element (mm) |
| $\psi, \psi_i$ | azimuth (°) |
| $r_i, r_o$ | radius of inner/outer raceway (mm) |
| $\psi_L$ | angular range of the load zone (°) |
| $R_i, R_o$ | radius of inner/outer groove (mm) |
| $Q_{\max}, Q(\psi)$ | maximum/contact force (N) |
| $c_{eq}$ | damping coefficient (Ns/m) |
| $P_d$ | radial clearance (mm) |
| $\alpha$ | contact angle (°) |
| $x, \dot{x}, \ddot{x}$ | displacement, velocity, acceleration of the outer or inner race (m, m/s, m/s$^2$) |
| $\beta_0$ | initial angle of the fault location (°) |
| $Z$ | number of rolling elements |
| $\Delta\beta$ | central angle (°) |

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
