# Peer review of "Dynamics Modeling and Analysis of Rolling Bearings Variable Stiffness System with Local Faults"

_machines, doi:10.3390/machines11060609_

Round 1

Reviewer 1 Report

This work as a whole is presented in sufficient detail: all the formulas necessary for modeling are given. Since there are many formulas, the part with the results is difficult to agree with the formulas. The authors should specify which formulas are used in the calculations. For example, Figure 10 shows the dependence of stiffness on angle. It is necessary to explain how this result follows from the formulas given.

For a better understanding of the work, it is necessary to make a separate list with notations in the article.

None of the figures show the main variables: theta and x.
As a rule, in mechanics, the variable x describes a change relative to some equilibrium position. It is necessary to explain what exactly equation (29) describes. And how does it agree with the theta angle?

Reviewer 2 Report

Authors studied the change of effective contact stiffness, contact deformation, contact force and total effective stiffness of rolling elements of bearings. They developed a dynamic model of a single degree of freedom with a single local damage fault and performed a simulation analysis and experimental verification.

Comments for improving the paper:

The state of the art should be extended at least with 10 references (ie: https://doi.org/10.1155/2014/2623 , https://doi.org/10.1016/j.protcy.2014.08.057,...).

The experimental verification: please, extend the comparison and stablish a numerical percentage for defining the coherence.

Authors should add a new section for discussing the results with other prediction methods. Also use new added references for discussing results.

In conclusion section, authors should add the limitations of the study and future works. Are authors thinking about applying the model to other bearing BBDD?

Reviewer 3 Report

This paper presents a method for dynamic modeling of the rolling bearing variable stiffness system with local faults, which provides an additional toolset for assisting the bearing fault diagnosis. The paper overall is in good shape. Some minor issues are listed below to further improve the paper's quality.

The Introduction needs to be strengthened. Machine learning based upon experimentally acquired signals has become the mainstream of fault diagnosis. Some recent works below also utilized the CWRU bearing dataset for the showcase. Another important class of methods for fault diagnosis is signal processing and feature extraction. Authors are suggested to discuss these facets briefly and show the motivation of this study, i.e., why they want to contribute by looking into the physical aspect.

10.1109/ACCESS.2020.2990528

https://doi.org/10.1016/j.measurement.2020.108502

https://doi.org/10.1007/s00170-021-08392-6

https://doi.org/10.1016/j.engappai.2021.104295

The methodology part includes a lot of mathematical equations. More references should be given if these equations were not originally proposed by the authors. Also, it seems that reference [17] is the main resource of the authors’ work. Authors are supposed to clarify the difference between their work and reference [17].

Some contact force spikes in Figures 11 and 12 were explained and noted, and some were not. It is helpful to explain all of them.

Is it possible to put the theoretical and experimental results together to highlight the agreement and thus validate the efficacy of the proposed method? Maybe adding another table to compare the key information extracted from the theoretical analysis and experimental investigation, respectively.

Proof read the manuscript and correct the errors if needed

Round 2

Reviewer 2 Report

Authors have adressed most the comments, but the paper can be improved taking into account these comments:

Although authors have clearly compared the results, however, I recommend to stablish a percentage for valuating the average coherence.

Authors should add the limitations of the study in conclusions section.

Reviewer 3 Report

NA

NA

Author Response

请参阅附件。
